# Incidence and predictors of mortality among patients with head injury admitted to Hawassa University Comprehensive Specialized Hospital, Southern Ethiopia: A retrospective follow-up study

**Desalegn Dawit Assele**[1]*, **Tigabu Addisu Lendado**[2], **Merid Assefa Awato**[2], **Shimelash Bitew Workie**[3], **Wolde Facha Faltamo**[3]

**1** Department of nursing, Hawassa University Comprehensive Specialized Hospital, Hawassa, Ethiopia, **2** Wolaita Zone Health Department, Wolaita Sodo, Ethiopia, **3** Department of Epidemiology and Biostatistics, School of Public Health, College of Health Sciences and Medicine, Wolaita Sodo University, Wolaita Sodo, Ethiopia

\* desalegndawit96@gmail.com

## Abstract

### Introduction

Head injury is the leading cause of morbidity and mortality throughout the world, especially in resource-limited countries including Ethiopia. However, little is known about the mortality rate and its predictors among these patients in Ethiopia. Thus, the study aims to assess the incidence rate of mortality and its predictors among patients with head injury admitted at Hawassa University Comprehensive Specialized Hospital.

### Methods

Institutional based retrospective follow-up study was conducted among 1220 randomly selected head injury patients admitted from July 2017 to July 2019. Bivariable and multivariable Cox regression models were fitted to identify the predictors of mortality. Proportionality assumption was tested by a global test based on the Schoenfeld residuals test.

### Results

The incidence of the mortality rate was 2.26 (95%CI: 1.9–2.6) per 100-person day observation. The independent predictors of time to death were age above 65 years (AHR:3.49, 95% CI:1.63, 7.48), severe TBI (AHR: 8.8, 95%CI:5.13, 15.0), moderate TBI (AHR:3, 95% CI:1.73,5.31), hypotension (AHR:1.72, 95%CI: 1.11,2.66), hypoxia (AHR:1.92, 95%CI: 1.33,2.76), hyperthermia (AHR:1.8, 95%CI: 1.23,2.63) and hypoglycemic (AHR:1.94, 95% CI: 1.34, 2.81) positively associated with mortality, while underwent neurosurgery was negatively associated with mortality (AHR: 0.25, 95% CI: 0.11,0.53).

**Data Availability Statement:** All relevant data are within the paper and its Supporting information files.

**Funding:** The author(s) received no specific funding for this work.

**Competing interests:** No authors have competing interests.

**Abbreviations:** AHR, Adjusted Hazard Ratio; GCS, Glasgow Coma Scale; ICP, Intracranial Pressure; ICU, Intensive Care Unit; TBI, Traumatic Brain Injury; IDR, Incidence density rate; PDO, person day observation.

## Conclusion

The incidence of mortality rate among head injury patients was high. Older age, moderate and severe TBI, hypotension and hypoxia at admission, neurosurgical procedure, and the episode of hyperthermia and hypoglycemia during hospitalization were the independent predictors of mortality among head injury patients. Therefore, intervention to reduce earlier deaths should focus on the prevention of secondary brain insults.

## Introduction

Head injuries are the most common cause of a neurosurgical emergency and continue to be a major cause of morbidity and mortality throughout the world. It is estimated that 69 million people suffer from Traumatic Brain Injury (TBI) from all cases per year [1]. Current estimates suggest that about 4.48 million people lose their lives due to injuries which accounts for 8% of all deaths globally, 38% more than the number of fatalities from malaria, tuberculosis, and HIV/AIDS combined [2]. Of these, an estimated 2 million deaths were attributed to the TBI [3], and the burden was concentrated in developing countries due to limited access to advanced life-sustaining measures after trauma [4].

In low-and middle-income countries, head injury patients have worse outcomes than patients in high-income countries [5]. Several studies in Africa have found that death rates from head injury range from 4.2% to 35% [6–9]. Ethiopia is not different from other developing countries; over 20% of head-injured patients die during treatment [10,11]. Evidence suggests that the possible causes for the high mortality rate could be older age, male gender, low GCS, and cause of injuries are likely non-modifiable risk factors and hypoxia, hypotension, hyperthermia, hypo or hyperglycemia, and did not undergo surgery or poor adherence to management guidelines are possible modifiable risk factors [12–14].

Clinical presentation of patients and advanced rescue care by emergency teams are crucial factors to determining favorable outcomes. Accordingly, non-surgical management should emphases on rapid transportation, avoiding hypotension and hypoxia, hyperthermia, and medical management to reduce brain swelling [15].

The management of TBI patients should be followed intracranial pressure monitoring is suggested to reduce post-traumatic death in the hospital within two weeks [16]. Unfortunately, in Ethiopia, prehospital care is not well established and hospitals are not well equipped [11]; this can increase the risk of secondary brain injury due to hypotension and hypoxia.

The government of Ethiopia has been striving to attenuate its devastating effects and to meet the sustainable development goal by halving RTIs related death. A previous study in Ethiopia did not address some significant predictors of mortality, and the contribution of censored subjects was not considered in determining predictors of mortality in head-injured patients [17]. This information is crucial for the mitigation of brain injury mortality. Therefore, this study aimed to assess the incidence rate of mortality and its predictors among head injury patients admitted at Hawassa University Comprehensive Specialized Hospital

## Methods

### Study design and setting

An institutional-based retrospective follow-up study was conducted at Hawassa University Comprehensive Specialized Hospital, from August to September 2020. Founds in Hawassa

City, 273 Km far south from Addis Ababa, Ethiopia. Since 2005, the hospital provides medical, surgical, ICU, and emergency care, out and in-patient service for about 18 million populations from the South region and other neighboring regions within 38 different departments. The hospital has 400 inpatient beds and 8 intensive care beds with 3 functional ventilators. The emergency department of the hospital provides services to 102,033 patients a year. In addition, this unit serves 1800 injured patients a year, including 960 head injuries [18].

## Population

All admitted patients presented between July 2017 and July 2019 with a history of head injury and associated impaired consciousness or radiographic evidence of traumatic brain injury. The study population was randomly selected from eligible head trauma patients admitted to HUCSH from July 8, 2017, to July 7, 2019. A total of 1,886 head injury patients was admitted to this hospital between July 8, 2017, and July 7, 2019. All head injury patients admitted to Hawassa University Comprehensive Specialized Hospital with a complete medical record in the period from July 2017 to July 2019 were included in the study. While incomplete patient records for baseline clinical features (GCS, BP, POS2), the outcome variable and follow-up time of fewer than 24 hours were excluded from the study.

## Sample size and sampling technique

The sample size was calculated based on double population formula by Epi info considering the following assumptions: 95% CI, power 80%, a ratio of unexposed to exposed 1:1, anticipated proportion of mortality rate among patients who did not undergo neurosurgery (unexposed) = 13.7% and AHR = 0.62 [19]. Therefore, the sample size of the exposed group is 610, and for the unexposed group 610, which yields 1220. Stata version 15 was used to generate a random sample. Then, using the subsequent unique number from the database registry, a random sample of 1,220 patients' cards was selected.

## Data collection tools and procedures

The English version data extraction checklist was developed from the literature [5,9,12]. The data were extracted from the patient's records, electronic database and, operation notes using a data extraction tool for the occurrence of the event. The checklist consists of demographic, injury characteristics, clinical and laboratory findings, and patient management variables. Vital signs and Glasgow Coma Score extraction was started at the time of admission and then following at every 6 hours base. The data were collected by five trained BSc Nurses by using ODK collect.

## Data quality management

To ensure the data quality, training was given for the data collectors and the supervisor about the ways of extracting the data based on the study objectives. The tool was also tested and the data were checked for consistency and completeness daily by the supervisor and principal investigator. Reviewed cards were boldly marked to avoid re-review. Data cleaning was checked by codebook for any missing values and data errors.

## Definition and operationalization of variables

The primary outcome variable is the death of head injury patients, the time variable is the time to occurrence of death measured from admission to date of the event and coded as one (death), zero for censored. Censoring: right censoring, are these cases like transfer-out, exist

with medical advice and lost to follow-up and patients alive at the end of follow-up. The predictor variables assessed were demographic factors (age and sex) and baseline clinical characteristics and laboratory factors (admission GCS, BP, temperature, RBS, HR, oxygenation, and seizure), and neurosurgery, and cause of injury.

Traumatic brain injury (TBI)–is an alteration in brain function which is manifesting as confusion, altered level of consciousness, coma, or seizure temporary or permanent following insult to the brain. Admission GCS was coded into mild (GCS of 13–15), moderate (GCS of 9–12), and severe (GCS of 3–8). Hypotension was defined as systolic blood pressure <90 mmHg measured at any time point during hospitalization, including the hospital arrival value. In pediatrics, hypotension is defined as SBP below the fifth percentile for age. This is estimated using the formula [70 mmHg + (age × 2)] [20,21]. Hypoxia was defined as a pulse oximeter oxygen saturation <90% at any time point during hospitalization, including hospital arrival value. Hyperthermia was defined as a temperature >38˚c during hospitalization [22]. Hypoglycemia was defined as a blood glucose concentration level <80mg/dL during hospitalization [22]

## Data processing and analysis

Data were collected by using ODK collect version 1.25.2 and then exported to STATA version 15 for analysis. Descriptive statistics including proportions, median, tables, and charts was done to describe the characteristics of the study participants. The Kaplan–Meier survival curve together with the log-rank test was fitted to test for the presence of difference in the incidence of death among the groups. The incidence of the mortality rate was calculated.

Confounding and effect modification was checked by looking at regression coefficient change if greater than or equal to 20% and multi-collinearity was checked using variance inflation factor and value of <10 was used as a cutoff point, indicating no collinearity [23].Variables significant at p-value< 0.2 level in the bivariable analysis were candidates for the final Cox regression analysis to identify the independent predictors of mortality.

Association was summarized by using an adjusted hazard ratio (AHR), and statistical significance tested was done at 95% CI and variables having a p-value less than 0.05 in the multivariable Cox proportional hazards regression model was considered as statistically significant. Proportionality assumption was tested by a global test based on Schoenfeld residuals. The goodness of fit of the final model was checked by Nelson–Aalen cumulative hazard function against Cox–Snell residual test [23].

## Ethical considerations

Ethical clearance was obtained from the ethical review committee of the College of Health Sciences and medicine of Wolaita Sodo University, and permission was obtained from Hawassa University Comprehensive Specialized Hospital. As the study was conducted through a review of patient records, no consent was obtained from the patients. Information about specific personal identifiers like the patient names were not collected and the personal informant was kept confidential throughout the study process.

## Results

### Demographic and injury-related characteristics of patients

A total of 1,220 records of head injury patients was reviewed. Of these, 1159 (95%) charts were included in the analysis, and the remaining 61 (5%) were excluded due to the incompleteness of baseline characteristics and outcome. The majority of patients were male 967 (83.43%).

**Table 1. Baseline demographic and injury-related characteristic head injury patients admitted to HUCSH from July 2017- July 2019, Hawassa, Southern Ethiopia.**

| Variables | Category | Death (N = 148) | Censored (N = 1,011) | PDO | IDR |
|---|---|---|---|---|---|
| Sex | Male | 126(13.03) | 841(86.97) | 5516 | 2.28 |
| | Female | 22(11.46) | 170(88.54) | 1026 | 2.14 |
| Age | < = 18 | 27(10.55) | 229(89.45) | 1655 | 1.63 |
| | 19—40 | 95(13.27) | 621(86.73) | 3891 | 2.44 |
| | 41—64 | 15(11.03) | 121(88.97) | 712 | 2.10 |
| | 65+ | 11(21.57) | 40(78.43) | 284 | 3.87 |
| Cause of injury | RTI | 113(15.96) | 595(84.04) | 4472 | 2.52 |
| | Fall down | 15(9.32) | 146(90.68) | 739 | 2.00 |
| | Assault | 20(6.90) | 270(93.10) | 1331 | 1.50 |
| Type of RTI (n = 708) | Pedestrian | 30(15.63) | 162(84.38) | 1211 | 2.47 |
| | Motorbike | 52(12.65) | 359(87.35) | 2496 | 2.08 |
| | Car | 25(23.81) | 80(76.19) | 780 | 3.20 |
| Polytrauma | Yes | 65(21.89) | 232(78.11) | 1831 | 3.54 |
| | No | 83(9.63) | 779(90.37) | 4711 | 1.76 |

Two-thirds, 716 (61.78%) of patients were between 19 and 40 years, and 51 (4.4%) of patients were above 65 years with a median age of 26 years (IQR: 20–35 years). Road accidents were the leading cause of head injury 708 (60.74%) and followed by assaults 282 (24.33%). A quarter of patients, 297 (25.63%) suffered multiple injuries to the extremities, chest, abdomen, pelvis, or spinal cord. The median time that patients arrived at the hospital after an injury was 6 hours (IQR: 2–20 hours). The overall median length of stay was 3 days (IQR: 2–6) (Table 1)

## Clinical and radiologic related findings

At admission, the median GCS score was 14 (IQR, 10–15). Seven hundred and sixty-nine patients (66.35%) had sustained mild TBI, a moderate TBI was 237 patients (20.45%), and a severe TBI was 153 patients (13.2%). Patients with a GCS of 3–8 had a mortality rate of 56.86%, as compared with 16.46% for those with a GCS of 12–9 and 2.86% for 13–15. At admission, 59 (5.09%) had hypotensive, 121 (10.4%) had hypoxic, and 96 (9.64%) had anemic. The majority of the patients, 909 (78.43%) had a head CT-scan done and 665 (73.16%) of the patients had an abnormal CT-scan. Of these, 185 (27.82%) were skull fractures and followed by 160, (24.06%), 142 (21.35%) were contusions and epidural hematomas, respectively. Regarding the pupillary light reaction, 98 (8.46%) patients with bilateral non-reactive and 92 (7.94%) had asymmetric pupils (Table 2).

## Management-related findings

Nine hundred thirty-seven 937 patients (80.85%), 131 (11.30%), 8 (4.4%) and were admitted to the casualty department, surgical ward, intensive care unit (ICU) and assisted by a mechanical ventilator, and neurosurgical ward, respectively. Over a third of the patients, third 374 (32.27) and 853 (73.6%) patients had received mannitol and phenytoin as prophylaxis of antiseizure, respectively. Among the cohort, 180 (15.53%) of the patients had undergone neurosurgery. Seventy-seven (42.78%) of patients underwent a craniotomy to evacuate the acute epidural hematoma and 62 (34.44%) of patients underwent exploratory burr holes. Of the 180 TBI surgeries performed, patients with mild TBI received the highest proportion (126, 70%), followed by patients with moderate TBI (34, 18.89%), and those with severe TBI (20, 11.11%). The mortality rate for all patients who underwent neurosurgery was 4.44% and 14.3% for those without neurosurgery (Table 2).

**Table 2. Baseline clinical and management related characteristics of head injury patients admitted to HUCSH from July 2017- July 2019, Hawassa, southern Ethiopia.**

| Variables | Category | Death (N = 148) | Censored (N = 1,011) | PDO | IDR |
|---|---|---|---|---|---|
| Hypotension | Absent | 114(10.36) | 986(89.64) | 6144 | 1.85 |
| | Present | 34(57.63) | 25(42.37) | 398 | 8.54 |
| Hypoxia | Absent | 83(8.00) | 955(92.00) | 5644 | 1.47 |
| | Present | 65(53.72) | 56(46.28) | 898 | 7.23 |
| Anemia (n = 996) | Absent | 104 (11.56) | 796(88.44) | 4866 | 2.13 |
| | Present | 28(29.17) | 68 (70.83) | 869 | 3.22 |
| Admission GCS | Mild | 22(2.86) | 747(97.14) | 3642 | 0.6 |
| | Moderate | 39(16.46) | 195(83.54) | 1590 | 2.45 |
| | Severe | 87(56.86) | 65(43.14) | 1310 | 6.64 |
| Pupillary light reaction | Bilateral reactive | 52(5.37) | 917(94.63) | 4987 | 1.0 |
| | Unilateral reactive | 37(37.76) | 61 (62.24) | 813 | 4.55 |
| | Bilateral nonreactive | 59(64.13) | 33(35.87) | 742 | 7.95 |
| Disposition | Emergency | 115(12.27) | 822(87.73) | 4046 | 2.84 |
| | ICU | 24(50.00) | 24(50.00) | 836 | 2.87 |
| | Surgical | 8(6.11) | 123(93.89) | 1174 | 0.6 |
| | Neurosurgical | 1(2.33) | 42(97.67) | 486 | 0.2 |
| Seizure | Present | 59(26.58) | 163(73.42) | 1631 | 3.61 |
| | Absent | 89(9.50) | 848(90.50) | 4911 | 1.81 |
| Hyperthermia | Absent | 71(7.11) | 927(92.89) | 4719 | 1.5 |
| | Present | 77(47.83) | 84(52.17) | 1823 | 4.22 |
| Hypoglycemia | Absent | 99(9.57) | 936(90.43) | 5818 | 1.7 |
| | Present | 49(39.52) | 75(60.48) | 724 | 6.77 |
| CT-scan (n = 1066) | Yes | 110(12.10) | 799(87.90) | 5688 | 1.93 |
| | No | 27(17.20) | 130(82.80) | 564 | 4.78 |
| CT-scan finding (n = 909) | Normal | 10(4.10) | 234(95.90) | 1236 | 0.8 |
| | Abnormal | 100(15.04) | 565(84.96) | 4452 | 2.24 |
| CT- scan pathology (n = 655) | Skull fracture | 6(3.24) | 179(96.76) | 1174 | 0.5 |
| | Epidural hematoma | 11(7.75) | 131(92.25) | 950 | 1.15 |
| | Subdural hematoma | 21(26.92) | 57(73.08) | 488 | 4.30 |
| | SAH | 4(40) | 6(60.00) | 96 | 4.16 |
| | ICH | 31(59.62) | 21(40.38) | 342 | 9.06 |
| | Contusion | 18(11.25) | 142(88.75) | 1184 | 1.52 |
| | DAI | 5(22.73) | 17(77.27) | 154 | 3.24 |
| | Others* | 4(25.00) | 12(75.00) | 64 | 6.2 |
| Neurosurgery | Yes | 8(4.44) | 172(95.56) | 1606 | 0.49 |
| | No | 140(14.3) | 839(85.70) | 4936 | 2.83 |
| Mannitol | Yes | 117(31.28) | 257(68.72) | 2885 | 4.05 |
| | No | 31(3.95) | 754 (96.05) | 3657 | 0.84 |
| Antiseizure | Yes | 135 (15.83) | 718(84.17) | 5142 | 2.62 |
| | No | 13(4.25) | 293(95.75) | 1400 | 0.92 |

Abbreviations: ICH: Intracranial hemorrhage, DAI: Diffuse axial injury, SAH: Subarachnoid hemorrhage, GCS: Glasgow Coma Scale, ICU: Intensive care unit. Others
* Pneumocephalus, maxillary and mastoid bone fracture.

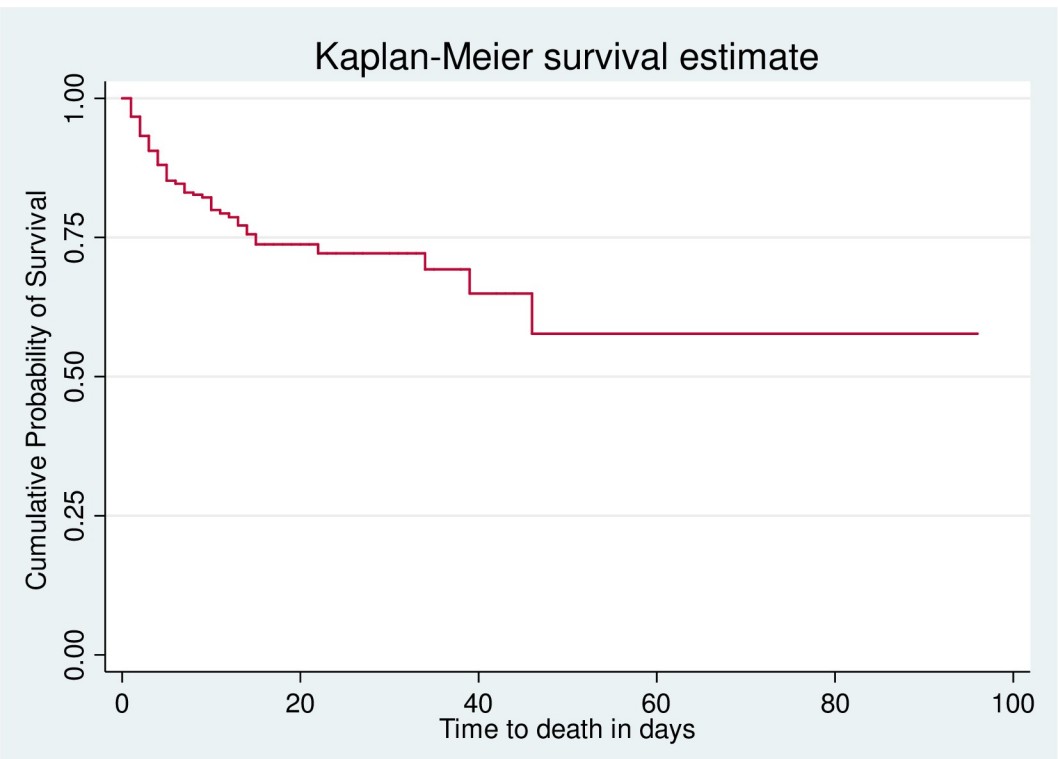

**Fig 1. Overall Kaplan-Meier estimation of their survival of admitted head injury patients in HUCSH, Hawassa, Southern Ethiopia; from July 2017 to July 2019.**

### Incidence of mortality and survival status

A total of 1,159 patients were followed for a minimum of 1 day and a maximum of 96 days, with a median follow-up time of 3 days. During the follow up period, 148[12.7% (95% CI = 10.9–14.8%)] were died, 904 [78% (95%CI = 75.5–80.2)] were cured and discharged, 19 (1.64%) were transferred out to other hospitals, 49 (4.23%) left against medical advice and 39 (3.36%) were lost from the follow-up.

The total time at risk for 1159 was 6,542 person-days of with an incidence rate of 2.26 (95% CI: 1.9–2.6) per 100-person day observation. The cumulative proportion of survival at the end of the first, second, seventh, 14th, and 96th days was 95.9%, 92.1%, 81%, 73%, and 55.6%, respectively (Fig 1). Among patients who died during the follow-up period, 126 (85%) were males, half (50.6%) died within the 48 hours of admission and more than 3/4th died in the casualty department.

The incidence of mortality in patients with severe TBI was 6.64 per 100 person-days it was 2.45 and 0.6 per 100 person-days for moderate and mild TBI, respectively. At three days of follow-up, the probability of survival for those sustained mild, moderate, and severe TBI was 98.1%, 92.5%, and 67.5%, respectively (Fig 2).

The incidence of mortality was lower among patients who underwent neurosurgery compared to those who did not undergo neurosurgery (Fig 3).

Kaplan–Meier survival curve together with the log-rank test was fitted to test for the presence of a difference in the occurrence of death among the categorical explanatory variables. There was a high incidence of mortality among patients with hypotension (log-rank test, $\chi2$ 84.53), hypoxia (log-rank test, $\chi2$ 144.21) and hyperthermia (log-rank test, $\chi2$ 100.42) (Table 3)

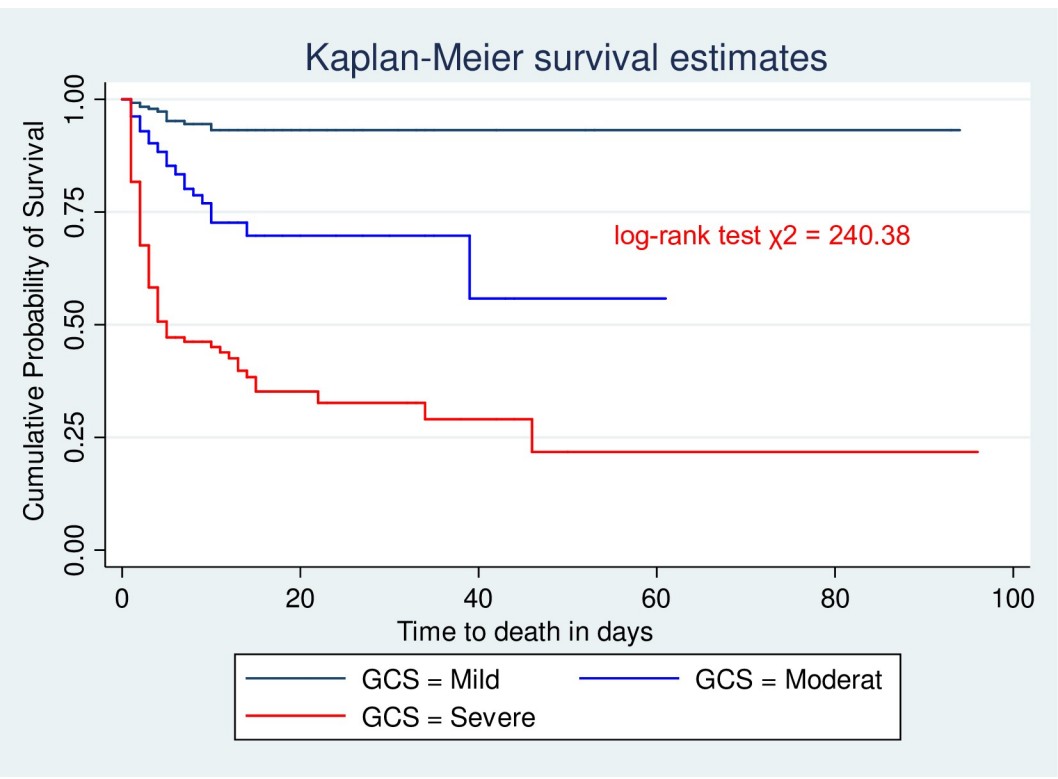

**Fig 2. Kaplan-Meier survival estimate for initial GCS among head injury admitted to HUCSH, Hawassa, Southern Ethiopia; from July 2017 to July 2019.**

### Predictors of death

In the bivariable analysis of selected baseline clinical and management variables (initial GCS, admission systolic BP< 90 mmHg, oxygen saturation < 90%, and neurosurgery), age, cause of injury, episode of seizure, episode of hypoglycemia, and episode hyperthermia were showed association with time to death at a p-value < 0.2. For clinical significance and control of confounding sex was included in the multivariate analysis. However, in the final multivariable Cox-proportional hazard model; age, initial GCS, admission systolic BP, oxygenation, neurosurgery, cause of injury, episode hypoglycemia, and episode hyperthermia were found to be independent predictors of death.

The risk of death among patients who underwent neurosurgery was 75% (AHR: 0.25; 95% CI; 0.11–0.53) times lower than those who did not undergo neurosurgery. The risk of death among patients with severe and moderate TBI was 8.8 (AHR: 8.8; 95% CI;5.13–15.0) and 3 (AHR:3; 95% CI; 1.73–5.31) higher than those with mild TBI, respectively. The risk of death among patients above 65 years old was 3.48 times (AHR: 3.49; 95% CI;1.63–7.48) higher than those under 18 years.

The risk of death among hypoxic patients was 1.92 (AHR: 1.92;95% CI;1.33–2.76) times higher than those who hadn't hypoxic. The risk of death among hypotensive patients was 1.72 (AHR:1.72[95% CI; 1.11–2.63]) times higher than those with normotensive. The risk of death among hypoglycemic patients was 1.94 (AHR: 1.94; 95% CI; 1.34–2.81) times higher than those with normoglycemic. The risk of death among hyperthermic patients was 1.8 (AHR: 1.8; 95% CI = 1.23–2.63) times higher than normothermic patients (Table 4).

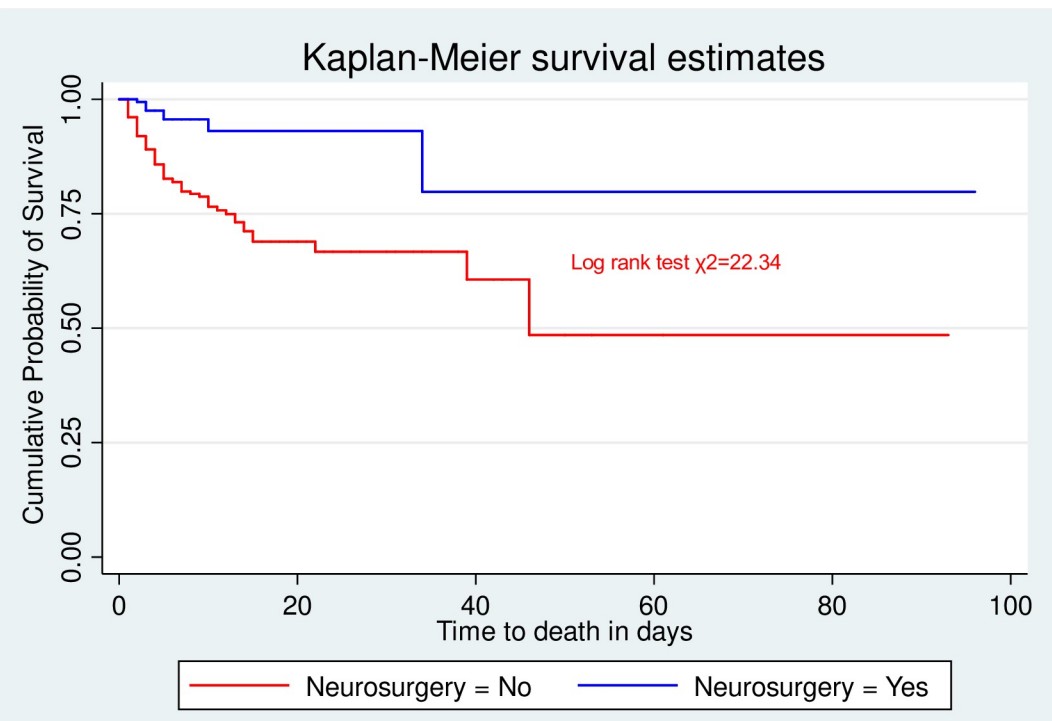

**Fig 3. Kaplan-Meier survival estimate for neurosurgery among head injury admitted to HUCSH, Hawassa, Southern Ethiopia; from July 2017 to July 2019.**

### Assessment of model adequacy

The overall Schoenfeld global test of the full model was checked for proportional hazard (PH) assumption and it was met (p-value = 0.783). All covariates are met the proportional-hazard assumption. Residuals were checked using the goodness-of-fit test by Cox Snell residuals. For the residual test, it was possible to conclude that the final model fits the data well (Fig 4). This figure shows if the Cox regression model fits the data, these residuals should have a standard censored exponential distribution with a hazard ratio. The hazard function follows the 45° line very closely.

### Discussion

This study found that 148 (12.7%) of patients had died in the follow-up period. This mortality rate is comparable to a study done in Rwanda 12.8% [19], the finding was lower than a study done in Iran 16.9% [24], in Nigeria 22.6% [25], and in Tanzania 30.7% [9].

**Table 3. Comparisons of death among different levels of predictor variables using the log-rank test for head injury patients to HUCSH from July 2017–July 2019, Hawassa, southern Ethiopia.**

| Variables | Log-rank ($\chi 2$) | p-value | Variables | Log-rank ($\chi 2$) | p-value |
|---|---|---|---|---|---|
| Sex | 0.18 | 0.678 | GCS | 240.38 | < 0.001 |
| Hypotension | 84.53 | < 0.001 | MOI | 4.13 | 0.1267 |
| Hypoxic | 144.21 | < 0.001 | Age | 5.42 | 0.1434 |
| Neurosurgery | 22.34 | < 0.001 | Seizure | 27.93 | < 0.001 |
| Hypoglycemia | 77.65 | < 0.001 | Hyperthermia | 100.42 | < 0.001 |

**Table 4. Bivariate and multivariate Cox regression analysis for independent predators of time to death among patients with head injury to HUCSH from July 2017-July 2019, Southern Ethiopia.**

| Variables | Category | CHR 95%CI | AHR 95%CI |
|---|---|---|---|
| Sex | Female | 1 | 1 |
| | Male | 1.01(.70, 1.73) | 0.9 (0.56, 1.44) |
| Age | < = 18 | 1 | 1 |
| | 19–40 | 1.42 (.92, 2.18) | 1.56 (0.99, 2.45) |
| | 41–64 | 1.18 (.63, 2.22) | 1.35 (0.74, 2.58) |
| | 65+ | 2.1 (1.04, 4.26) | 3.49(1.63, 7.48) * |
| MOI | Fall down | 1 | 1 |
| | RTI | 1.50(.87, 2.58) | 1.12(.63, 1.97) |
| | Assault | 0.74 (.37, 1.44) | 1.89(.92, 3.86) |
| Hypotension | Absent | 1 | 1 |
| | Present | 4.9(3.38, 7.31) | 1.72(1.11, 2.66) * |
| Hypoxia | Absent | 1 | 1 |
| | Present | 5.7(4.15, 7.98) | 1.92(1.33,2.76) ** |
| Admission GCS | Mild | 1 | 1 |
| | Moderate | 4.5 (2.72, 7.76) | 3(1.73, 5.31) ** |
| | Severe | 16 (10.2, 26.2) | 8.8(5.13, 15.0) ** |
| Neurosurgery | Yes | 0.21 (0.10, 0.43) | 0.25(0.11, 0.53) ** |
| | No | 1 | 1 |
| Hyperthermia | Present | 4.6(3.33, 6.46) | 1.80(1.23, 2.63) * |
| | Absent | 1 | 1 |
| Hypoglycemia | Present | 4.07 (2.88, 5.74) | 1.94(1.34, 2.81) ** |
| | Absent | 1 | 1 |
| Seizure | Present | 2.3(1.68, 3.27) | 1.13(0.79, 1.63) |
| | Absent | 1 | 1 |

Abbreviations: GCS, Glasgow coma scale, CHR, crude hazard ratio; CI, confidence interval; AHR, adjusted hazard ratio, MOI, mechanism of injury

* significant at a value <0.05 level

** significant at a p value<0.001 level.

However, the finding is higher than a study done in Tanzania 9% [26], in Rwanda 9.3% [12], and Uganda 9.6% [14]. This high mortality rate could be the result of a lack of prehospital care, scarcity of ventilators and ICU beds, limited surgical capacity, and poor adherence to protocols.

Neurosurgical intervention was found to be a preventive association with mortality. This finding is consistent with other studies [19,27,28], the possible reason could be an evacuation of extra-axial hematoma, which helps to manage increased intracranial pressure, decreased the risk of death and permanent neurologic deficit by preventing secondary brain injury due to herniation of brain [16]. The evidence suggests that timely evacuation of the hematoma has reduced the risk of mortality and length of stay [29]. However, only 20 (11.1%) of severe TBI patients underwent surgery, which might bias the estimate. These may be patients with a lower GCS and advanced age, who are not eligible as neurosurgical candidates. Patients with moderate TBI receiving surgery was shown that the greatest improvement in treatment outcome was followed by mild and severe TBI [27].

The risk of death among patients older than 65 years was higher than those younger than 18 years. This finding is in agreement with other studies [19,30,31], the possible reason could be for these older were more likely to have a low initial GCS, depressed immunity, less likely to adequately respond to the treatment, and increased risk of infection.

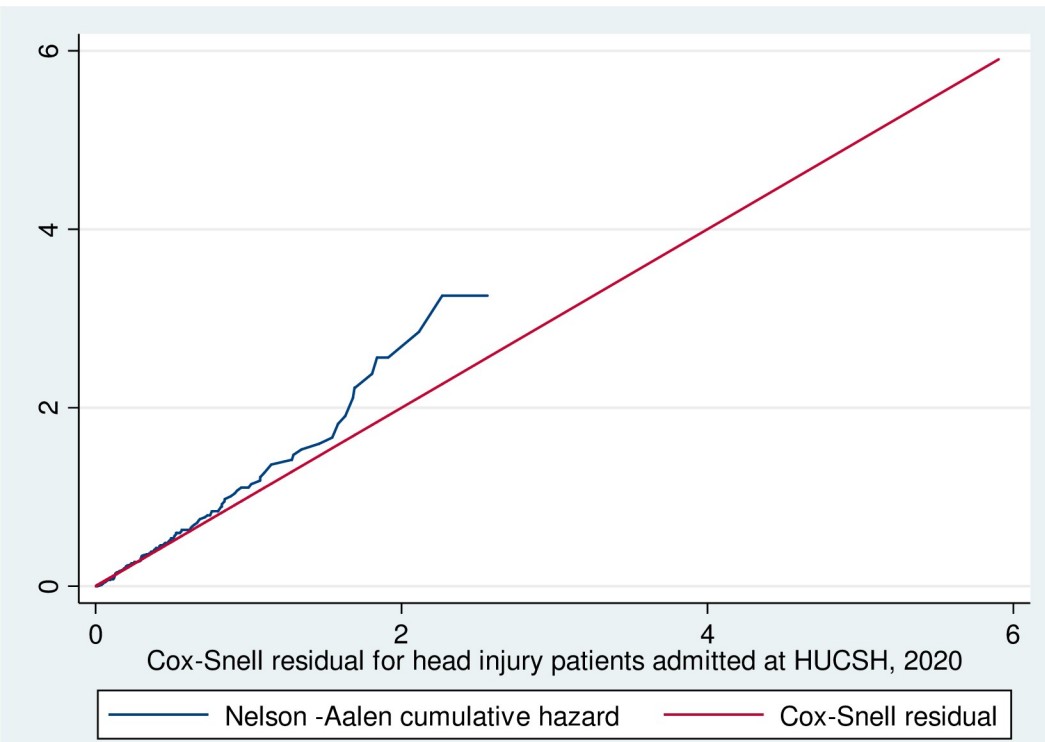

**Fig 4. Nelson-Aalen cumulative hazard graph against Cox-Snell residual on head injury patients at HUCSH, 2020.**

The risk of death among patients with severe and moderate TBI were 9 times and 3 times higher hazard of death as compared to mild TBI, respectively. This finding was in agreement with other studies [19,27], the possible reason could be airway impatience, high risk of infection, not a candidate for surgery, and not assisted with a ventilator and poor adherence to management guidelines. Therefore, care providers should implement and adhere to guidelines, that can reduce the risk of mortality and permanent disability [32,33].

The risk of death for hypotensive patients was 1.72 times the increased hazard of death as compared to normotensive patients. This finding is supported by different studies [24,32,34], but in other, hypotension was not a predictor of mortality and neurological outcomes [5]. This could be due to the study period and design, sample size, and algorithm for triage and treatment of trauma patients in the emergency department.

The risk of death among patients with hypoxia had 1.92 times higher as compared with patients without hypoxia. This finding was consistent with the previous studies done in Tikur Anbessa Specialized Hospital [17] and University Teaching Hospital of Kigali [12]. Besides, the risk of dying for a patient who needs mechanical ventilation was 4 times higher than haven't need for mechanical ventilation [31]. This could be due to the restricted flow of oxygen to the brain, increased metabolic demand, and hypercapnia leads to vasodilatation that could increase an intracranial hemorrhage [35,36].

Accordingly, hypoglycemic patients were 1.94 times higher hazard of death as compared to normoglycemic patients. This finding was in line with the previous study [37], this could be explained by the fact that hypoglycemia leads to deprivation of the brain of its fuel which can lead to compromised brain energy metabolism [35]. Furthermore, the hazard rate of death among patients who were exposed to hyperthermia was 1.8 times higher as compared with normothermic patients. This finding is in agreement with another study [38], this could be since hyperthermia increases brain metabolic demand, ICP, and cell damage [39].

## Limitation of the study

The findings of this study might suffer from the fact that it is a retrospective study and based on records; some variables were missing, while the others were not recordable. The study failed to track the deaths that occurred at home and this may underestimate the mortality rate because patients discharged with medical against were at high risk of complications and death. Moreover, since the majority of the observations were censored, there may be a potential bias due to excluded records and the unknown status of absconders.

## Conclusion

The incidence of mortality was higher in head injury patients admitted to Hawassa University Comprehensive Specialized Hospital, with half of all deaths occurring 48 hours after admission. Admission GCS, hypotension, hypoxia, age above 65 years, did not undergo neurosurgery, episode of hyperthermia, and hypoglycemia during hospitalization were the independent predictors of mortality.

## Recommendations

Care providers give special emphasis should be paid to patients with impatient airways, low GCS, hypoxia, and hypotension at admission, and should follow patients' vital signs attentively and adhere to management guidelines. The Ministry of Health and stakeholders should improve the capacity of neurosurgical care, improve the critical care capacity and skilled human power are recommended to increase the survival of patients with head injuries. Furthermore, further study using the prospective study design would better compensate for the limitations of this study.

## Supporting information

**S1 Data.**
(DTA)

## Acknowledgments

We would like to say thanks to Wolalita Sodo University college of health sciences and medicine for facilitating this study. Special thanks go to the medical record office staff at Hawassa University Comprehensive Specialized Hospital for their valuable information. Lastly, our gratitude goes to data collectors for their kind and excellent cooperation during data collection.

## Author Contributions

**Conceptualization:** Desalegn Dawit Assele, Wolde Facha Faltamo.

**Data curation:** Desalegn Dawit Assele, Shimelash Bitew Workie.

**Formal analysis:** Desalegn Dawit Assele.

**Funding acquisition:** Desalegn Dawit Assele.

**Investigation:** Desalegn Dawit Assele.

**Methodology:** Desalegn Dawit Assele, Wolde Facha Faltamo.

**Project administration:** Desalegn Dawit Assele.

**Resources:** Desalegn Dawit Assele.

**Software:** Desalegn Dawit Assele.

**Supervision:** Desalegn Dawit Assele, Tigabu Addisu Lendado, Merid Assefa Awato.

**Validation:** Desalegn Dawit Assele.

**Visualization:** Desalegn Dawit Assele.

**Writing – original draft:** Desalegn Dawit Assele, Tigabu Addisu Lendado, Merid Assefa Awato, Shimelash Bitew Workie, Wolde Facha Faltamo.

**Writing – review & editing:** Desalegn Dawit Assele, Tigabu Addisu Lendado, Merid Assefa Awato, Shimelash Bitew Workie, Wolde Facha Faltamo.

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
