## [Decision Letter · Decision Letter 0]

10 May 2021

PONE-D-21-10673

Incidence and Predictors of Mortality among patients with head injury admitted to Hawassa University Comprehensive Specialized Hospital in southern Ethiopia, 2020

PLOS ONE

Dear Dr. Assele,

Thank you for submitting your manuscript to PLOS ONE. After careful consideration, we feel that it has merit but does not fully meet PLOS ONE’s publication criteria as it currently stands. Therefore, we invite you to submit a revised version of the manuscript that addresses the points raised during the review process.

First of all: Your manuscript is not written in standatd english. Please, get help from a professional native english speaker. Reviewer 1 has contributed many helpful recommendations to improve your manuscript. Follow his instructions.

We look forward to receiving your revised manuscript.

Kind regards,

Hans-Peter Simmen, M.D., Professor of Surgery

Academic Editor

PLOS ONE

Journal Requirements:

2. Thank you for including your ethics statement: "Ethical clearance was obtained from the ethical review committee of the College of Health Sciences of Wolaita Sodo University, and permission was obtained from Hawassa University Comprehensive Specialized Hospital. "

3. PLOS ONE does not copy edit accepted manuscripts (https://journals.plos.org/plosone/s/criteria-for-publication#loc-5). To that effect, please ensure that your submission is free of typos and grammatical errors.

5. Please amend either the title on the online submission form (via Edit Submission) or the title in the manuscript so that they are identical.

6. Please ensure that you refer to Figures 1-3 in your text as, if accepted, production will need this reference to link the reader to the figure.

7. Please upload a copy of Figure 4, to which you refer in your text. If the figure is no longer to be included as part of the submission please remove all reference to it within the text.

Reviewers' comments:

Reviewer's Responses to Questions

**Comments to the Author**

1. Is the manuscript technically sound, and do the data support the conclusions?

Reviewer #1: Partly

Reviewer #2: Partly

2. Has the statistical analysis been performed appropriately and rigorously? 

Reviewer #1: Yes

Reviewer #2: I Don't Know

3. Have the authors made all data underlying the findings in their manuscript fully available?

Reviewer #1: Yes

Reviewer #2: No

4. Is the manuscript presented in an intelligible fashion and written in standard English?

Reviewer #1: Yes

Reviewer #2: No

5. Review Comments to the Author

Reviewer #1: I read your work with great interest. At present TBI is one of the most important topics when dealing with trauma-patients all around the world.

Here I have gathered a few general remarks regarding your manuscript:

- When using the term "polytrauma" you do not give the definition used at your institution to classify patients as polytraumatised. Since the definition in use can have relevant influence on the prognosis of the treated patients you should clarify this issue.

- You do not mention the time from trauma to start of treatment of your patients nor do you give the amount of time they suffered from conditions like hypoxaemia, hypothermia etc. - this would be of great importance to put your findings into perspective.

- In your section "Incidence of mortality and survival status" you report on the overall follow-up of all of your patients but do not give the follow-up regarding e.g. different severities of TBI. Such a division into subgroups would greatly improve the relevance of your manuscript.

In this section I also found the following sentence: "(Error! Reference source not found.)". Please clarify!

In the same section you state "patients who had underwent surgery had higher mean survival time when compared to those did not undergone surgery (sic)". But you do not go into detail regarding the underlying pathologies as well as the characteristics of patients receiving operative or conservative treatment. You rightly mention this flaw in your discussion but do not give any further details.

- The allover quality of your manuscript would benefit greatly from proofreading by an english native speaker (as one of many possible examples I would like to mention the use of "causality" in stead of - I suppose - "casualty" in your sections about "Management related findings" and "Incidence of mortality and survival status" or the sentence "The risk of death hyperthermic patients was 1.8 (AHR: 1.8; 95% CI= 1.23– 2.63) times higher than those with normothermic patients (Table 4)." found in your "Assessment of model adequacy").

Only after thorough revisions of your manuscript I would deem it eligible for PLOS ONE.

Reviewer #2: Dear Authors,

thank you very much for submitting for manuscript. Doing research to improve patients' treatment and outcomes especially in the presence of limited resources is very impressive and you have collected a lot of important data on your patients. However I cannot recommend publication of the manuscript in its current form.

- Language of the manuscript needs significant improvement to make it more understandable

- Variables should be shown in a separate table

- Data on follow-up should be added

- Why did you limit the patient sample to 1220 if you had more patients?

- Secondary injury being a major contributor of bad outcomes has long been known.

6. PLOS authors have the option to publish the peer review history of their article (what does this mean?). If published, this will include your full peer review and any attached files.

Reviewer #1: No

Reviewer #2: No

---

## [Author Response · Author response to Decision Letter 0]

15 Jun 2021

Thank you for your inspiring comment.

---

## [Editor Report · Decision Letter 1]

23 Jun 2021

Incidence and predictors of mortality among patients with head injury admitted to Hawassa University Comprehensive  Specialized Hospital, Southern Ethiopia: A retrospective follow-up study

PONE-D-21-10673R1

Dear Dr. Assele,

We’re pleased to inform you that your manuscript has been judged scientifically suitable for publication and will be formally accepted for publication once it meets all outstanding technical requirements.

Kind regards,

Hans-Peter Simmen, M.D., Professor of Surgery

Academic Editor

PLOS ONE
---

## [Editor Report · Acceptance letter]

11 Aug 2021

PONE-D-21-10673R1 

Incidence and predictors of mortality among patients with head injury admitted to Hawassa University Comprehensive Specialized Hospital, Southern Ethiopia: A retrospective follow-up study 

Dear Dr. Assele:

I'm pleased to inform you that your manuscript has been deemed suitable for publication in PLOS ONE. Congratulations! Your manuscript is now with our production department. 

Kind regards, 

on behalf of

Dr. Hans-Peter Simmen 

Academic Editor

PLOS ONE